# White Atmospheres: Choreographing Racial Materialities in Academic Space

**Ben Spatz** 

School of Arts and Humanities, University of Huddersfield, Huddersfield HD1 3DH, UK; b.spatz@hud.ac.uk

**Abstract:** This essay offers a critical introduction to the circulation of racial materialities, and especially whiteness, in North American and European academic contexts. It proposes that we can escape from the dominant epistemology of identity as a fixed attribute of individuals without losing the urgent and much-needed analytics of identity as social and material force. In the gap between "identity politics" and a richer critical politics of identity lies the difference between a discursive public sphere of agonistic conflict and one of potentially transformative relationality. Drawing on critical race theory and especially black radical thought, my analysis rejects the reduction of identity to discrete census categories and attempts to situate contemporary scholarly practices in the context of a planetary decolonial movement. If "identity" today is all too frequently dismissed by a methodological whiteness that strictly separates it from materiality, politics, and knowledge, then a dramaturgical or *choreographic* analytics of race might better address how racial materialities operate both above and below the level of individual bodies, subjects, and citizens. Synthesizing practical insights from artistic research and performing arts with critical theories of race and identity, this essay refers to some of the author's recent personal experiences at academic events in order to describe and analyze whiteness as a form of social choreography.

**Keywords:** practice research; artistic research; dramaturgy; choreographic practices; critical whiteness; critical race theory; black studies; social epistemology; poststructuralism; identity politics

How can we escape the dominant epistemology of "identity" as a fixed attribute of individuals, without losing the urgent and much-needed analytics of identity as social and material force? In the gap between these two approaches—we might say, between "identity politics" and the politics of identity—lies the difference between a discursive public sphere of agonistic conflict and one of meaningful and potentially transformative relationality. To put this in the terms of the critical race theory and black radical thought on which I will be drawing here, within this gap lies the difference between, on the one hand, a white-supremacist reduction of identity to atomized census categories that are misperceived as symmetrical and, on the other, a planetary decolonial movement in which the portal of identity might lead to a shared world of actual politics.[1] If "identity" today is all too frequently dismissed by a methodological whiteness that strictly separates it from what is legitimately political, then a dramaturgical or *choreographic* analytics of race and other sociocultural formations might better address how racial materialities operate both above and below the level of individual bodies and persons. Synthesizing practical insights from artistic research and critical theory, this essay builds on the premise that racial (and other related) identity formations are as present, material, and real in spaces of practice as anything else. Racial identities in this sense are intrinsic to the social choreography of academic events and inseparable from their internal structures.

In a recent book chapter called "We Have Always Been Queer", Diana Taylor describes a series of events that took place at the Hemispheric Institute's 2014 Encuentro in Montreal. As she writes, "a very queer dispute erupted in the assembly of some eight hundred participants" when controversy over a stage performance led to political fragmentation and conflict between the "self-identified WE's" who had come together for the

conference event.[2] I similarly wish to discuss and analyze a handful of events in which I have recently participated, wherein political fragmentation and conflict arose around questions of identity and especially whiteness. Like Taylor, I will conclude my discussion by reflecting on a particularly intense Long Table session in which these dynamics became palpable in time and space. The Long Table is a social choreographic structure, developed by Lois Weaver, that "disrupts hierarchical notions of expertise" and can be used "to invite community knowledge around difficult conversations."[3] Improving in several ways upon the conventional roundtable and moderated discussion formats, the Long Table also carries its own risks in being unmoderated and hence potentially enabling greater conflict to occur. In approaching the technical and organizational structures of academic events as a social choreography of racial and other identities, I am drawing on intersections of critical race theory, black studies, and practice-based artistic research, as found for example in the critical dramaturgical account of Ralph Lemon's choreographic work offered by Katherine Profeta.[4] Following a recent wave of brilliant and incisive work in black studies, I figure identities here as complex ontological entities that cannot be reduced to the quantifying or empirical methods of conventional social and historical research.

It is beyond the scope of this essay to conduct a review of how blackness and intersectional identity are theorized in those works. Rather, I take inspiration from them here primarily to rearticulate the modes and materialities by which dominating and hegemonic forms of cultural-material whiteness structure many academic events. Among the diverse touchstones of critical black studies that have influenced my thinking are those produced by Marquis Bey, Denise Ferreira da Silva, Tiffany Lethabo King, Katherine McKittrick, Fred Moten and Stefano Harney, Joshua Myers, Jennifer Nash, Damani Partridge, Cedric Robinson, Christina Sharpe, Alexander Weheliye, and Frank Wilderson. While there are many crucial differences between these interventions, both in their arguments and in their disciplinary positioning, as a body of work they have succeeded in substantially shifting the terms by which blackness is critically theorized, putting forward a range of new frameworks that resolve or refuse to separate what have conventionally been distinguished as psychological, social, and material dimensions of racialization. In my view, a necessary response to this work has not yet been developed to rethink whiteness.[5] I therefore begin here to unpack the material presence and circulation of whiteness in terms that exceed such disciplinary distinctions. Instead of dividing whiteness into its psychological, social, and material components, I assume that methodological whiteness is part of what constructs those divisions and that a different kind of critical grip is therefore needed to understand it. For this reason, I also do not provide a singular definition of whiteness, but instead offer a practice-based account of its social choreographic materialities as I have come to recognize them in particular situations. My aim here is not to offer a comprehensive framework for analyzing racial materialities through a choreographic lens, but merely to take some initial steps towards the integration of critical race theory and black studies into the formulation of social choreography in predominantly white spaces.

A small conference in 2017 was perhaps the first occasion when I took the chance to publicly name the whiteness of an academic event while it was happening. I do not remember the exact context in which I referred to the predominant whiteness of the institutional and disciplinary context in which we were gathered. I only remember the way in which one audience member, whose exclusion from the visual domain of whiteness was obvious, approached me afterwards, his evident relief mixing with near disbelief: "Did you say… whiteness?" he asked me, almost incredulous. Indeed, I had named whiteness in a hegemonically white space, a space in which whiteness was not (and almost could not) be named. Since that time, I have continued to name whiteness more and more. I have tested these waters by more or less gently suggesting that a decolonial framework informed by recent black studies might better enable scholar-practitioners to understand some of the recurrent problems of epistemology, ecology, and sexuality that are more often explicitly thematized in research, especially in Europe. In doing so, I have also become increasingly aware of how conversations about whiteness can crash and fail when they are

reduced to questions about who is or is not white and what that means on an individual level. I have therefore tried more and more strictly to avoid making assumptions about any given individual's racial politics, understanding, or identity, while *at the same time* more and more vocally directing attention towards these exact phenomena on levels of analysis other than that of the individual. In other words, I more and more explicitly and frequently name "whiteness" as the hegemonic context and structure of the academic events I attend, while at the same time studiously avoiding what I would call the crucial misstep of piling responsibility for those structures onto the presumed identities of individuals. This approach will in some ways be familiar to critical race thinkers, but it can be deepened and expanded through the embodied and artistic methods that I am calling, in the context of this special issue, social choreography.

As a first example, we might foreground the racial dimension of a ubiquitous yet rarely examined structuring element of academic events: the timetable. Nearly every academic conference, seminar, workshop, or symposium is organized and structured on a basic level by a written program, often in both digital and print media, that lays out a timetable with associated information about the names and identities of participants alongside the titles and brief descriptions of their sessions and presentations. Biographical sketches and session abstracts may offer rich critical content, but the skeleton of the event program is a formal schedule formulated in the logic of clock time. At a major international conference, this program can run to hundreds of pages, a thick book that is printed for the occasion and then archived. Smaller conferences may have elegant program booklets or just a sheet of paper listing the schedule. And nearly all formal academic meetings have written agendas, often including the approval of minutes from the previous meeting. All of these circulating documents are examples of what I have come to call "white writing": a particular form of writing that does not so much describe, analyze, or interpret but instead lays out explicit rules and quantified formulae to determine what will or should happen. Like religious commandments, constitutional law, species taxonomies, notated musical meter, and clock time itself, the event timetable sets forth a written structure around which the practices of life are then more or less strictly organized.[6] Indeed, the atmospheric whiteness of any given event could be said to depend at least in part upon how strictly a written schedule determines what actually happens.

When I lived at the rural home base of a theatre company in Poland, our daily schedule rarely followed an explicit plan. As actors, we would often be called to arrive at the theatre in the early evening, but this did not mean that the ensemble's work would begin at that time. Instead, we actually began whenever the director arrived and stopped when he decided. This approach gave us an opportunity as actors to cultivate a sense of constant readiness and availability that was integral to our performance practice. An organic flow of time was sought that could involve longer or shorter sessions and various kinds of breaks, ending after just a few hours or pushing on until the early morning. Instead of following a predetermined schedule, the director acted as a conductor of time: a role that could involve great responsibility and care in the taking of risks or alternatively lapse into tyranny. More recently, I attended an academic event where I was surprised to discover on arrival that there was no schedule or program as such. In this case there was no charismatic or tyrannical director and the flow of time was much more casual and relaxed. I found myself disoriented, not because I was concerned that we would waste our time together, but because I found the social dynamics challenging. As a relative outsider to the group, I was not sure how much space to take up, when to speak, or how best to introduce myself. This experience could be taken as highlighting the value of explicit temporal structure. Alternatively, it may demonstrate the pervasiveness of white approaches to time and my own incapacity to navigate time otherwise.

The choreographic approach I want to suggest here could be a kind of alternative to the timetable as the primary way of translating an academic event into written words. Whereas a program is full of specific times ("coffee break at 11:00"; "plenary session from 2:30–4:00 p.m.") as well as the names, titles, and affiliations of individuals and the titles

and abstracts of papers and presentations, a different accounting might ask how whiteness and other identities are present and active in ways that slip below or above the radar of the formal program. If the schedule is a core element of structural whiteness, then so are the buildings and architectures in which such events most often take place. The types of chairs and tables that are available, the shape and lighting of the rooms, the signage, the relationships with cleaning staff and others who maintain the property, as well as the neoliberal university as landlord—all are material instantiations of whiteness that relentlessly choreograph the practice of gathering. Recently, to make this point, I picked up a plastic coffee cup that was sitting on a table during a para-academic gathering and said: "This cup is made of whiteness". The cup was not white in color; rather, my statement was intended to reject the division of racial identity and material politics according to which the ubiquity of a substance like plastic is seen as a neutral or purely economic rather than racial phenomenon. Plastic can be understood as a powerful form of whiteness in multiple ways. Like whiteness, it is increasingly ubiquitous, not only in the macro infrastructures of transportation, communication, architecture, and furnishings, but also as plastic microparticles that are increasingly found in all liquids, from the deep ocean to the human bloodstream. The attractiveness of plastic as a modern element is apparent. It is malleable, strong, and flexible; it can be used in so many different ways.[7] The undeniable usefulness of plastic is exceeded only by the increasing obviousness with which our living ecologies are choking to death on it. So it is with whiteness and white writing: ubiquitous, flexible, useful, deadly.

To track whiteness in this way is to recognize that the question of who is white and what that means cannot be isolated from much larger matters of power, knowledge, and structure. Of course, it will be necessary for individuals to face their own varying relationships to whiteness and for academic gatherings to create and support spaces in which these relationships can be analyzed. But if we do not first name the massively material whiteness of the institutional contexts in which we come together—not only the university, but also the city, the state, and the economy—then we will fail to grasp the radical asymmetries that structure every event and interaction there. Although vast literatures deal with structural racism, often their theories and languages substitute populations and statistics for critical engagements with daily forms of practice. As a result, it is only too easy for much implicitly white academic discourse to continue with business as usual, as if its very objects and methods of understanding were not fundamentally structured in racial ways. On the one hand, the politics of race and identity are excluded from the core work of research, as if disciplines like philosophy, media, or sociology were not built from the ground up on racial histories and forms. On the other hand, when the problem of race does become present and palpable in ways that cannot be so easily set aside, this may be experienced as an unfair attribution of the burden of colonial history to individuals classified as "white people." Neither of these framings is able to grapple effectively with the multiple ways in which whiteness circulates apart from skin color: as networks of wealth and kinship, as US American citizenship, as English-language fluency.

I continue to be surprised by the degree of defensiveness and fragility that some scholars display when I raise this topic. I am still encountering the argument that race and racialization are old ways of thinking from which we should move on into a self-evident liberal humanism of individuals. At a recent talk I gave on whiteness and artistic research, an anthropologist asked me directly why I was still thinking in terms of racial categories. He told me that he had been working collaboratively with communities of indigenous people in the Global South for decades and that their encounters unfold through direct human interaction, not through racial identities. I was not sure how to respond. I did not want to undercut what may well be very meaningful relationships developed over many years. At the same time, I cannot understand how such relationships could be truly collaborative or even mutually respectful without an understanding of the role that European colonialism continues to have as a structuring materiality on a global scale. Did this researcher not feel the pressing relevance of colonialism and its racial categories upon his own capacity to

travel and work with those people? Or would he acknowledge the force of colonialism, but refuse to accept that it operates in a structuring way within his own and others' daily embodiment, locating it solely in an abstract realm of laws and economies? Despite such impasses, I am gradually finding a sense of community amongst people who recognize their multilayered relationships to whiteness and who want to understand how whiteness is present, and how it might be dismantled, in our lives.

Although whiteness is deeply sedimented in the choreographies of embodiment, these may in some ways be more open to change and transformation than the large-scale architectures and infrastructures through which those bodies move. This is why I find it so urgent to avoid reducing whiteness to an attribute of individuals, while continuing to speak more and more about it as a structural and material force—not only in sociological terms but in the terms developed by critical humanities and especially in theatre, dance, and performance studies. Concepts like the dramaturgical and the choreographic are urgently needed to get beyond the methodological limitations of disciplines like sociology and law. To approach whiteness as a fixed attribute of bodies is precisely to accept a white methodology, a white epistemology that for the most part rigorously excludes artistic and embodied knowledges. We cannot break out of a census-style epistemology of identity by ignoring the prevailing categories of identification, but only by rearticulating them in yet stronger material terms. To think of whiteness as a choreography is precisely to understand that it is always both a matter of what bodies *are* and what bodies *do*. While legal and sociological methodologies tend to fix the identities of individuals in order to count them, a social choreographic approach foregrounds the interplay of being and doing: a body becomes white through white actions, techniques, and knowledges. While the plasticity of embodied practice should not be overstated, it is indeed more malleable than the literally plastic and concrete infrastructures that previously enacted choreographies have laid down for us as the inheritence of the modern world.

Whiteness in this pervasive, atmospheric sense is so powerful in structuring practice that we could think of it as another kind of gravitational axis. Gravity defines a clear vertical axis perpendicular to the earth, but the strictness of this axis (which results from the earth being many orders of magnitude more massive than the objects on its surface) evidently does not mean that objects can only move in straight lines up and down. In the same way, while proximity to whiteness may define a strictly linear axis of power within white institutional space, this does not mean that actions and identities can only move in a straight line either toward or away from whiteness. Consider how many ways there are to deal with the verticality of gravity: one uses gravity to throw a ball in an arc, to jump and dance, to walk along a road, to roll horizontally on wheels. In the same way, the vertical axis of proximity to whiteness affords many different kinds of movement, strategies, and techniques that are not simply reducible to aligning oneself with or against hegemonic power. One does not get far by pretending that gravity or institutional whiteness does not exist. On the contrary, one constantly has to align oneself with whiteness in order to get anything done. As with physical gravity, one must recognize and understand the dominant axis in order to be able to calculate one's actions so as to move sideways, or even (temporarily) straight up. Although one may wish to act against whiteness, one always does so having negotiated a more or less strategic position. Since we are talking here about movement in relation to power rather than to physics, this is precisely a matter of social choreography: how we position and move ourselves given the pervasive, structuring presence of forces that are far more durable and powerful than our bodies.

The choreography of relations to whiteness is complex and differs according to what other forces we are able to activate. While Kimberlé Crenshaw's famous metaphor of intersectionality between identity positions applies well to legal and sociological frameworks, a performance-based dramaturgical or choreographic methodology allows us to think in more dynamically temporal, spatial, and intercorporeal ways about how these different forces might interact in a given moment. To take an especially poignant example in the present moment: to activate jewishness in an academic context—as social choreography

rather than a religious, national, or ethnic category—requires very careful navigation of the gravitational force of whiteness. Indeed, the reason why the political meaning of jewishness, Zionism, and the ongoing genocide in Palestine/Israel are such major focal points globally is because they function as symbols for the kinds of strategic and contingent relationships to whiteness that structure many if not all politicized identities today. In the moment that the meaning of jewishness becomes bound to the question of the legitimacy of the state of Israel, everyone who might wish to activate the former as a critical social force is compelled to take a position in relation to global whiteness and coloniality: for or against? More nuanced positionalities are sidelined, and the complexity of jewish history and identity rendered apparently superfluous, as the space becomes intensely polarized according to a singular axis of whiteness. Diasporic jewishness may be a particularly stark example of such polarization, but a similar phenomenon applies also to the forces that gather under names like feminism, queerness, disability, and even other quasi-racial namings such as indigeneity, brownness, and the designation "of color." From a strictly critical position, it is evident that such activations can align themselves either with whiteness (as in white feminism, homonormativity, and normatively white disability politics) or against it. But a social choreographic perspective recognizes that such alignments are rarely as simple or obvious as movement up or down along a vertical axis. Rather, the atmospheric and material forces of institutional whiteness demand that our navigations through white space be full of eddies, spirals, and tangents. As much as one may wish to position jewishness, feminism, queerness, disability, or other such materialities against the global frame of white coloniality, this "againstness" will always consist of a specific navigation of forces at a given time and place.

If whiteness is atmospheric in this sense, then there is no absolute top or bottom to its spatiality, just as there is no top or bottom to the gravitational axis—only relative movement up and down, towards and away from power. The "other" to whiteness, its negative or earthly pole, is then not an alternative agonistic formation of cohesive power but a dynamic web of non-nationalist sovereignties. To operate counter to whiteness, or even to begin to unpack one's own complicities with it, requires a recognition of the complexities and complicities through which one relates to whiteness—as well as ongoing embodied, artistic, and social choreographic research into specific techniques of disidentification from whiteness, which may range from fugitivity to sabotage to overt conflict. The opposite of a white-supremacist colonial nation-state, for example, is not a differently ethnic or racialized nation-state, but an alternative, nonstate political formation and an approach to sovereignty that is ontologically distinct from the white concept of property.[8] Similarly, the strategies and techniques by which white academic institutionality can be negotiated range from the "quiet quitting" of escape and sabotage to declarations of postdisciplinarity and the abolition of the university itself. These distinctions can be choreographed and perceived in relation to jewishness, feminism, disability, and other identity formations, but they are perhaps most important to recognize and cultivate in relation to blackness as the formal racial opposite of whiteness.

Given that racial opposition, and the inherent antiblackness of white space, the appearance of blackness as a social choreographic force in such spaces will be of particular importance. Again, this can never be reducible to the presence of individuals, as might be suggested by a narrowly demographic approach to identity. Understanding blackness as an attribute of individual bodies puts far too much pressure on the people to whom it is attached, as well as erasing the manifold ways in which molecular or atmospheric blackness circulates and transforms. This is a precise analogy, in social choreographic terms, of the point just made about nation-states: to foreground the relationship of individuals to blackness as a counter or antidote to whiteness is to reduce blackness to an individualist ontology that is actually definitive of whiteness, which constructs itself both historically and in the present through the figure of the individual, the logocentrically coherent citizen, erasing all other forms of being and relation. Instead, the field of critical black studies offers far richer and more generative formulations of social choreographics, in its multiple

articulations of blackness beyond the demographic. Blackness as formulated by the authors mentioned above is a social choreography of racialized materialities that radically exceeds the ways in which we have been taught to name and describe "race" as an individual identity. The task of social choreography, then, might be to develop new techniques, both critical and artistic, by which to activate formations like the queer, the feminist, the disabled, and the brown, within a field of social materialities defined by blackness in that sense.

Having described academic institutional space as structurally and atmospherically white (materially structured by a gravitational axis of whiteness), shot through by other identity formations moving in complex choreographic ways, and comprehensible only through a serious engagement with critical black studies, I will conclude as stated with a brief account of a recent Long Table session during which the topic of whiteness was explicitly thematized. According to the official instructions for Long Table sessions: "No one will moderate / But a host may assist you." On this particular occasion, the Long Table was the final session of a symposium in which the potential for conflict around race and whiteness had arisen. No substantial interpersonal conflict had yet erupted, as far as I could tell, yet the possibility was there. To avoid harm, and to take responsibility for having convened this particular group, I decided that I needed to push the role of "host" and the notion of "assistance" towards that of moderation. Although I did not formally moderate, I broke with my usual practice—setting up a Long Table and then leaving it for others to gather around—by staying at the table for the entire two hours. Internally I felt that, like the host of a dinner party (the inspiration for the Long Table format), it was my duty to remain present until I believed that everyone at the table felt safe. Since that moment never came, I did not leave the table until the session formally ended.

It was a powerful experience for me to embody the role of host in such a way. I found myself intensively activated on a bodily level, with a heightened awareness of moment-to-moment interactions that could almost be compared to what I have experienced in dance and theatre performance and performer training. I believed it was my responsibility to honor and respect everyone who was present, while at the same time refusing to fall back into a premise of apparent neutrality that could only reinscribe the hegemonic whiteness of the institutional context. In other words, I found myself to be literally dancing with, alongside, and against whiteness in the context of this academic conversation. What took place inside me during the session was precisely an internal social choreography. I was particularly aware of my own relationships to whiteness, blackness, and jewishness in that moment, although my queerness and nonbinary gender, my minor disability and "small fat" embodiment, my dance and theatre training, and my childhood family dynamics were all present too. As I have suggested in this essay, none of these identifications—not even, or especially not, relations to whiteness and to blackness—can be adequately grasped as distinct or symmetrical attributes of a person or body. Rather, the ways in which they interacted within me and through my actions were radically asymmetrical in a sense that can be usefully called choreographic. Together, these internal and external forces determined how I sat at the table and how I interacted with the others who joined me there. With whom did I make eye contact? At whom did I smile? When did I look around the room at those witnessing the conversation? For whom did I try to create additional time and space to speak and whom, on rare occasions, did I interrupt? All of these micro actions and behaviors, which occur routinely in everyday life, were magnified in that moment precisely because of how the hidden architecture of structural whiteness had been named. It is always there, but this was a moment in which researchers from different contexts, materially formed by different identifications and differently positioned in relation to academic institutionality, were attempting to navigate it differently.

After the Long Table ended, I gave thanks to those who had helped me organize and host the symposium, which was then formally concluded. I do not remember who called out in that moment: "Ben should start a new university!" This was an exciting idea, but also a strange responsibility to attach to me, when I had merely served as host of the event. I immediately recalled, perhaps to counterbalance the idealism of that remark,

a question that had been posed to me by another participant the night before, during a private conversation about the atmospheric whiteness of the space: "Could there be more people of color here?" This question was also directed at me, although it was not explicitly stated that it would have been my job to effect this change. I could take such a question as an indication of my failure as host, just as I could take the call for me to start a new university as an indication of my success. But I would prefer to decenter myself from both comments, foregrounding instead the racial atmospheres in which we work and the social choreographies through which we navigate them. My response to both remarks is therefore the same: Perhaps, but none of this can be accomplished at the level of individual action. Can we work—can we choreograph this—together?

**Funding:** This research was funded by University of Huddersfield, grant number ICF200-07.

**Data Availability Statement:** The data are contained within the article.

**Conflicts of Interest:** The author declares no conflict of interest.

## Notes

1  In this essay, I lowercase identity terms such as black, jewish, and european to indicate their irreducibility to discrete, census-style categories. A fuller explanation of this choice, including a much longer discussion of dance dramaturgy as a theoretical framework for the materialization of race in practice, can be found in Spatz (2024).

2  See (Taylor 2020), p. 153.

3  See (Weaver 2003).

4  See (Profeta 2015).

5  For example, Sarah Ahmed's phenomenological approach to whiteness (Ahmed 2006, 2007) has not been integrated into the more socially and historically oriented critical whiteness studies of Linda Martín Alcoff (2018). The more recent work of Nicholas Mirzoeff (2023) begins to move in this direction, importantly through an engagement with visual and artistic production that has also been central to black studies.

6  On the circulation of policy documents in academia, see (Ahmed 2012). On musical meter as a colonial imposition, see (Robinson 2020). For the concept of white writing, see Spatz (2024). The role of timetabling in European history, from the daily schedule of monastic life through the temporal structuring of armies and schools, is a central interest in the work of Michel Foucault.

7  For more on the racial history of plasticity as a concept, see (Thakkar 2024).

8  On nonstate political formations, see (Boyarin 2023; Anderson 2021). On alternative meanings of sovereignty, see (Teves et al. 2015).

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
