# Peer review of "White Atmospheres: Choreographing Racial Materialities in Academic Space"

_arts, 2023_

Round 1

Reviewer 1 Report

Comments and Suggestions for Authors

The essay asks an essential question: what are the material ways whiteness structures scholarly meetings and discussions?  Further, the essay turns to choreography and dance to offer a series of analytics to help illuminate what the piece terms the "atmospherics" of academic spaces and conversations, shaping how participants interact. The stakes of this essay are clear and important to the kind of knowledge that can be produced.

I do not think, however, that this piece is ready for publication at this time. A fairly significant revision would help it achieve its aims. I offer some suggestions to improve the piece:

- The piece never fully defines what it means by whiteness, identity, choreography, structure and never fully works through the binaries it relies on. It is clear the author is gesturing toward several ongoing scholarly conversations, but those debates are nuanced, with scholars frequently defining terms in competing ways. The footnote about sovereignty and the one that lumps Frank Wilderson with the Combahee River Collective don't do justice to the vigorous debates these terms have engendered. This reader would benefit from a more direct effort at defining terms. Perhaps, more choices need to be made to clarify or narrow the focus.

- I was excited by the idea that dance theory had much to offer CRT theories from law and sociology. Focusing more narrowly on this angle would have interested this reader. It would require a more systematic engagement with law, sociology, and dance theory.

- The whiteness examined in this article seems more akin to Wilderson's Afropessimism than CRT's interest convergence or racial realism.  This, again, is an important debate to have, but this piece does not fully engage it. (Also, there have been huge debates about in the intersectionality wars, which are not really discussed or referenced but seem relevant here).

- The examples, two conference meetings the author attended or read about, seem like insufficient evidence to make the kind of claim the author wants to make, especially since the author needs to keep certain things anonymous. For such an important theoretical point, the reader needs a bit more to hold onto. The claim about the whiteness of time (along with the implication about the ownership of ideas and social status at conferences) are all worthy of articles in and of themselves. Further, the author would need to trace out their implications more fully than just the limited sketch offered here.

- The point about what to do with recordings from the long table was interesting but seemed out of place in this piece.

I like where this piece is going, but I don't think it achieves its aims yet. I would be eager to see another, more focused version of this essay/paper.

Comments on the Quality of English Language

I felt like the essay format worked against the argument being made. The tone blended informality and jargon. The wording blended registers. The sentences were lengthy, sometimes needlessly so, and frequently veered toward a more theoretical or philosophical piece. If the author wants to write an essay, then craft sentences more appropriate for that kind format.

Author Response

I have tried to respond to this reviewer's desire for a more focused essay. I cannot change the overall tone of the writing, which mixes essayistic personal account with the language of critical theory. My hope is that other readers will find the critical theoretical language more intriguing and engaging, rather than considering it "jargon."

There is not space for me here to define all of these terms adequately. I have added further signposting to my much more extensive discussion of closely related issues in a monograph that is being published next month. I have moved the main references to critical black studies into the main text (rather than a footnote) and have explained why I do not go further into this work and why I do not offer precise definitions of whiteness or identity here. I have added language to clarify that I am in no way "lumping" diverse voices in black studies together, but rather responding to the development of that field from my own perspective.

I think that the main concern of this reviewer has to do with the disciplinary framing of this essay, which is firmly grounded in current black studies and the critical humanities. I have not seen the table of contents for this special issue, but my hope is that the editor's introduction will provide further context, for example on the concept of "choreography," which will help frame my essay.

Similarly, I hope that the material I have added about critical whiteness studies will be sufficient for the reader to understand my overall approach. I think it is more clear now that I am not trying to use the events I discuss as "evidence" of the hegemonic and atmospheric whiteness of these academic spaces, which I take as a given based on decades of work in critical race studies. Rather, I am taking them up as simple case studies through which to work through what I consider to be the necessarily racial analytics of a social choreographic approach. This is more explicit in the current version.

One helpful suggestion was that the "point about what to do with recordings from the long table was interesting but seemed out of place in this piece." I have removed the discussion of audiovisual recording, which is very important in my larger project but probably too much to include here.

As I discussed with the special issue editor, I think this reviewer is looking for a different kind of writing: more empirical, with precise definitions and more scholarly apparatus. That is not the purpose of my essay, although much of it can be found in the monograph that I cite in note 1. I would therefore suggest that the revised version not be shared with this reviewer a second time.

Reviewer 2 Report

Comments and Suggestions for Authors

The article uses a few related but competing metaphors to interrogate the idea of whiteness: atmosphere, gravity, and molecules. First, the use of these different metaphors reduces the clarity of thought. (The use of the molecular metaphor seems in conflict with much of the articular that speaks to the influence of whiteness as broader than any individual or molecular understanding of the concept.) The article would retain the force of its argument with the removal of any reference to molecular. Additionally, the article uses some examples that only raise further questions about the article's thesis. (The example of the white plastic cup leaves the reader to determine if the plastic is representative of whiteness or only in the specificity of it being a white cup and what either understanding means for how the author makes a statement on whiteness.) Finally, there is an assumption that the reader understands terms like choreography as a modifier of relationships or dramaturgical. This assumption may be fair, depending on the journal's audience. For those interested in this discussion of race, these terms may need defining.

Author Response

This reviewer was more positive about my original draft and did not mark any criteria as "must be improved", implying that only minor revisions were needed.

Following their direct suggestion below, I have removed the use of the molecular metaphor from this essay. The molecular is a key concept in the monograph I cite in note 1, however I agree that I have insufficient space to introduce it here alongside the main discussion of choreography linked to the special issue topic.

With regard to how the "example of the white plastic cup leaves the reader to determine if the plastic is representative of whiteness or only in the specificity of it being a white cup," I have clarified explicitly now that the cup was not white in color at all and I have added further discussion of why I am linking whiteness to plastic as a material substance and force.

As with the other reviewer, there is a question of how the reader should understand the term "choreography" in this context. This reviewer notes that there may not be a problem here as long as the audience is familiar with an expanded sense of the choreographic beyond theatrical dance works. I would further assume that the editor's introduction to the special issue topic, as well as the other pieces in this special issue, will provide sufficient additional context for the notion of the choreographic. My aim here is not to advance the main theme of social choreography but to demonstrate how a specific kind of racial analysis (one deriving from critical black studies rather than, for example, sociology) ought to be applied in this context.

Round 2

Reviewer 1 Report

Comments and Suggestions for Authors

The author did a tremendous job responding to the feedback on the first draft. This revised version is a powerful, clear, and engaging. It offers a much clearer main claim and then fully explains how choreography and dramaturgy can offer the basis of a revised form of critical race theory. I loved the use of gravity as a metaphor to explain racial dynamics. 

The main piece of feedback I have is related to discussing time's relation to whiteness. It seems to me that what the piece means is "conception of time," not time itself. It seems like the same conception of time that structures the academic conference and the academy as a whole is rooted in European and captialist conceptions of linear time, which can be exchanged and imbued with social relations. I could see a snarky reader asking how is 2:00 pm an example of whiteness.  Offering a more nuanced framing of the conception of time might help.  The footnote to Foucault is getting at this point, but I do think the argument needs to be made in the body of the essay itself.

My second observation, not a critique, is that whiteness is pretty abstract in the piece . Some readers may need or want a clearer definition or set of examplesm, earlier in the piece.